# Repeated Social Defeat Enhances CaCl_2_-Induced Abdominal Aortic Aneurysm Expansion by Inhibiting the Early Fibrotic Response via the MAPK-MKP-1 Pathway

**DOI:** 10.3390/cells11040732

**Published:** 2022-02-19

**Authors:** Hiroshi Kubota, Hiroyuki Yamada, Takeshi Sugimoto, Naotoshi Wada, Shinichiro Motoyama, Makoto Saburi, Daisuke Miyawaki, Noriyuki Wakana, Daisuke Kami, Takehiro Ogata, Masakazu Ibi, Satoaki Matoba

**Affiliations:** 1Department of Cardiovascular Medicine, Graduate School of Medical Science, Kyoto Prefectural University of Medicine, Kyoto 602-8566, Japan; kbt-h@koto.kpu-m.ac.jp (H.K.); sugimoto@koto.kpu-m.ac.jp (T.S.); wada-n@koto.kpu-m.ac.jp (N.W.); motoyama@koto.kpu-m.ac.jp (S.M.); msaburi@koto.kpu-m.ac.jp (M.S.); torisan@koto.kpu-m.ac.jp (D.M.); nw0920@koto.kpu-m.ac.jp (N.W.); matoba@koto.kpu-m.ac.jp (S.M.); 2Department of Regenerative Medicine, Graduate School of Medical Science, Kyoto Prefectural University of Medicine, Kyoto 602-8566, Japan; dkami@koto.kpu-m.ac.jp; 3Department of Pathology and Cell Regulation, Graduate School of Medical Science, Kyoto Prefectural University of Medicine, Kyoto 602-8566, Japan; ogatat@koto.kpu-m.ac.jp; 4Department of Pharmacy, Kinjo Gakuin University, Nagoya 463-8521, Japan; ibi@kinjo-u.ac.jp

**Keywords:** psychosocial depression, cardiovascular disease, abdominal aortic aneurysm, perivascular fibrosis, wound healing, vascular smooth muscle cells, MAP, MKP-1

## Abstract

Depression is an independent risk factor for cardiovascular disease and is significantly associated with the prevalence of abdominal aortic aneurysm (AAA). We investigated the effect of repeated social defeat (RSD) on AAA development. Eight-week-old male wild-type mice were exposed to RSD by being housed with larger CD-1 mice in a shared cage. They were subjected to vigorous physical contact. After the confirmation of depressive-like behavior, calcium chloride was applied to the infrarenal aorta of the mice. At one week, AAA development was comparable between the defeated and control mice, without any differences being observed in the accumulated macrophages or in the matrix metalloproteinase activity. At two weeks, the maximum diameter and circumference of the aneurysm were significantly increased in the defeated mice, and a significant decrease in periaortic fibrosis was also observed. Consistently, the phosphorylation of the extracellular signal-regulated kinase and the incorporation of 5-bromo-2′-deoxyuridine in the primarily cultured aortic vascular smooth muscle cells were significantly reduced in the defeated mice, which was accompanied by a substantial increase in mitogen-activated protein kinase phosphatase-1 (MKP-1). The MKP-1 mRNA and protein expression levels during AAA were much higher in the defeated mice than they were in the control mice. Our findings demonstrate that RSD enhances AAA development by suppressing periaortic fibrosis after an acute inflammatory response and imply novel mechanisms that are associated with depression-related AAA development.

## 1. Introduction

Mental disorders are an important cause of debility worldwide and are a main contributor to comprehensive disease burden and represent substantial causes of death [1]. Meta-analyses have revealed that depression is related to a greater risk of cardiovascular disease and subsequent outcomes after myocardial infarction [2,3,4,5]. There is increasing evidence that depression is causally connected to the development of atherosclerotic cardiovascular disease (CVD) through the integration of various factors; however, the detailed mechanisms of depression-related CVD development remain to be fully elucidated [6,7,8].

Abdominal aortic aneurysm (AAA) is one of the most widespread atherosclerotic diseases and has a comparatively high mortality rate, particularly in elderly people presenting with rupture [9]. Wide-ranging struggles have been made to improve a therapeutic approach to prevent the expansion of AAA; however, an effective therapeutic method has not yet been established [10,11]. Until recently, less attention has been paid to the association between depression and AAA development compared to the attention paid to coronary heart disease and stroke, partially because of an inverse association between diabetes mellitus and incident AAA [12]. Daskalopoulou et al. were the first to indicate that past depression was more significantly associated with AAA incidence than newly onset depression [4]. Recently, a population-based prospective study showed that patients with depressive symptoms have a significantly higher risk of incident AAA after the adjustment of conventional risk factors [13]. Likewise, a nationwide cohort study showed that the prevalence of depression was significantly higher in patients with AAA, independent of treatment modality [14]. These findings suggest that depression is significantly related to the development of AAA than was previously thought and imply that an additional mechanism other than conventional cardiovascular risk factors is implicated in depression-related AAA development.

Tsankova et al. showed that the experience of repeated social defeat (RSD) resulted in the appearance of depressive-like behavior in a mice experiment [15]. We previously reported that RSD evokes depressive-like behavior in apoE^−/^^−^ mice and that it promotes atherosclerosis [16]. We recently showed that RSD promotes fibrin-rich clot formation by exaggerating the formation of extracellular neutrophil traps [17]. Here, we showed that RSD enhances AAA development after arterial damage in wild-type (WT) mice. Despite there being no difference in the acute inflammatory response after arterial injury, the periaortic fibrotic response was markedly decreased in defeated mice, which was accompanied by a lower accumulation of α-smooth muscle actin (SMA)-positive cells in the adventitia than in the control mice. Interestingly, primary cultured aortic vascular smooth muscle cells (VSMCs) from defeated mice showed that the platelet-derived growth factor (PDGF)-induced phosphorylation of the extracellular signal-regulated kinase (ERK) and the incorporation of 5-bromo-2′-deoxyuridine (BrdU) were markedly reduced, which was accompanied by the augmented expression of mitogen-activated protein kinase (MAPK) phosphatase-1 (MKP-1), an endogenous MAPK inhibitor. Our findings may provide novel insight into the relationship between depression and AAA development.

## 2. Materials and Methods

### 2.1. Repeated Social Defeat

Male wild-type (C57BL/6J) and CD-1 mice were obtained from Shimizu Laboratory Supplies Co., Ltd. (Kyoto, Japan) and were maintained on a normal diet (12.0% fat, 28.9% protein, 59.1% carbohydrate; Oriental Yeast Co., Tokyo, Japan). Mice that were 8- to 10-weeks old were subjected to repeated social defeat (RSD) as previously reported [18], with minor modifications. After screening the aptitude of the aggressor CD-1 mice, each CD-1 resident mouse was housed with an intruder mouse. The two mice were not in direct physical contact and were connected via a holed partition that allowed constant visual, auditory, and olfactory contact without physical contact. During the 10 consecutive days of stress induction, the intruder mice were placed into another resident’s home cage each day to avoid familiarization with the aggressor. The control mice were housed separately in the same type of cage and did not have any physical interaction with the CD-1 resident mice. The animals were housed in a room maintained at 22 °C under a 12 h light/dark cycle and were provided with drinking water ad libitum.

### 2.2. Behavior Analysis

Behavioral analysis was performed as described previously [19,20,21]. In the tail suspension test, the mice were held from their tails. The immobility time was recorded for 6 min using a charge-coupled device (CCD) video camera (JIN-608AC; Kyohritsu Electronic Industry Co., Ltd., Osaka, Japan). In the social interaction test, the time spent in the interaction zone when the target was absent or present was recorded using a CCD video camera. The social interaction ratio (SIR) was calculated by dividing the interaction time spent in the presence of the target by the time spent in the absence of the target. The RSD-exposed mice with an SIR less than 1.0 were certified as defeated mice with depression-like behavior, while non-exposed mice exhibiting an SIR that was not less than 1.0 were certified as control mice.

### 2.3. Mouse Aneurysm Model

After the behavioral analysis, AAAs were developed via the periaortic application of 0.5 m calcium chloride (CaCl_2_), as previously described [22]. Ten-week-old mice were anesthetized by isoflurane (2%, 0.2 mL/min) using an anesthetic instrument (PITa-Quark; Sanko Manufacturing Co., Ltd., Saitama, Japan) during the surgery. The depth of the anesthesia was examined via the lack of a tail pinch response and was carefully monitored during the surgery. A laparotomy was performed under sterile conditions with the support of a stereomicroscope (Leica Microsystems K.K., Tokyo, Japan). After the abdominal aorta between the left renal artery and iliac bifurcation was carefully exposed, CaCl_2_-treated gauze was attached to the surface of abdominal aorta for 15 min. The gauze was them removed, and the intraperitoneal cavity was thoroughly washed three times with 0.9% sodium chloride (NaCl). In the sham-operated mice, CaCl_2_ was substituted with 0.9% NaCl. For the MKP-1 inhibition experiment, BCI (Dual Specificity Protein Phosphatase 1/6 Inhibitor) (HY-115502; MedChemExpress, Monmouth Junction, NJ, USA) was intraperitonially injected twice (at day 3 and 10 after CaCl_2_ application) at the dose of 35 mg/kg in 100 μL of 10%DMSO, 40% PEG300, 5% Tween80, and 45% Saline, as previously described [23].

### 2.4. Hemodynamic Analysis

Under conscious and unrestricted conditions, blood pressure and heart rate were recorded using a sphygmomanometer (BP-98A; Softron, Tokyo, Japan). 

### 2.5. Serum Concentration of Corticosterone

Serum corticosterone concentration measurements were sourced from FUJIFILM Wako Pure Chemical Corporation, Osaka, Japan.

### 2.6. Aneurysm Measurement and Histological Analysis

Transcardial perfusion with 4% paraformaldehyde was performed under anesthesia. The abdominal aortic tissue was detached by removing the adjacent fatty and scar tissues to ensure that the aortic wall clearly discernible. After recording images, we measured the maximal diameter of aneurysmal aortas using ImageJ software v1.50i (https://imagej.nih.gov/ij/index.html, accessed on 20 November 2021). The abdominal aorta, including a part of the maximal diameter, was cut out and embedded in paraffin. Cross-sections of the aortic tissue were stained with Elastica van Gieson or Masson’s trichrome stain. The aneurysmal and non-aneurysmal portions of the aortic sections were defined according to the structural integrity of an elastic plate in Elastica van Gieson or Masson’s trichrome stain.

### 2.7. Immunohistochemical Analysis

Three consecutive sections were prepared from the middle portion of the maximal diameter of the AAA. For the F4/80 immunological staining, the anti-F4/80 antibody (1:100, ab6640; Abcam, Cambridge, UK) and Alexa Fluor 488-conjugated secondary antibody (Thermo Fisher Scientific, Waltham, MA, USA) were used. For the matrix metalloproteinase (MMP)-9 staining, the anti MMP-9 antibody (1:100, ab38898; Abcam) and Alexa Fluor 555-conjugated secondary antibody (Thermo Fisher Scientific) were used. The anti-α-SMA antibody (1:200, ab5694; Abcam), Alexa Fluor 488-conjugated secondary antibody (Thermo Fisher Scientific), and Alexa Fluor 555-conjugated secondary antibody (Thermo Fisher Scientific) were used. For MKP-1, the anti- MKP-1 antibody (1:200, sc-373841; Santa Cruz Biotechnology, Dallas, TX, USA) and Alexa Fluor 488-conjugated secondary antibody (Thermo Fisher Scientific) were used. The sections were examined using an LSM 510 META confocal microscope (Carl Zeiss, Jena, Germany). Non-immune immunoglobulin Rabbit IgG and polyclonal isotype control were used as negative controls for α-SMA and MMP-9. For the non-immune immunoglobulin Rat IgG2b, the kappa monoclonal isotype control was used as the negative control for F4/80, and for the nonimmune immunoglobulin mouse IgG2b, the kappa monoclonal isotype control was used as the negative control for MKP-1. Positive staining was analyzed using ImageJ software v1.50i (https://imagej.nih.gov/ij/index.html, accessed on 20 November 2021). The number of F4/80- and MMP-9-positive stained nuclei were assessed in two sections comprising six animals from each group. The percentages of the MKP-1-positive stained nuclei were assessed in two sections comprising six animals from the control group and eight animals from the defeat group. The percentages of the MKP-1-positive stained nuclei in the α-SMA-positive stained nuclei were assessed in two sections comprising four animals form the control group and four animals from the defeat group.

### 2.8. Quantitative Real-Time Polymerase Chain Reaction (qPCR)

The total RNA content was extracted from the abdominal aortic tissue using the RNeasy Fibrous Tissue Mini Kit (74704; Qiagen, Hilden, Germany) and was reverse transcribed to prepare cDNA with the TAKARA Prime Script RT reagent Kit with gDNA Eraser (RR047A; Takara Bio, Shiga, Japan). Real-time PCR was performed using a CFX384 Touch Real-Time PCR System (Bio-Rad Laboratories, Inc., Hercules, CA, USA) with a KAPA SYBR^®^ FAST Universal qPCR Kit (KK4602; KAPA Biosystems, Wilmington, MA, USA). The data are shown as gene expression levels relative to those of the controls. The primer pairs are listed in the Appendix A.

### 2.9. Ex Vivo MMP Activity

Ex vivo matrix metalloproteinase (MMP) activity was examined using an in vivo imaging system (IVIS), as previously described [24]. An amount of 2 nmol MMPSense 750 FAST (Perkin Elmer, Boston, MA, USA) was injected via the tail vein. The abdominal aorta was excised 6 h after injection, followed by ex vivo aorta imaging using an IVIS Lumina Series III optical imaging platform (PerkinElmer Inc.) while adjusting the red filter (excitation, 749 nm; emission, 775 nm long pass). Regions of interest (ROIs) encircling the abdominal aorta were drawn by hand, and the certifying signal was evaluated in units of scaled counts per second.

### 2.10. Primary Culture of VSMCs in the Thoracic Aorta

The vascular smooth muscle cells (VSMCs) of the descending thoracic aorta were prepared from the 8-week-old control and defeated mice as previously described [25] because the amount of tissue in the infrarenal aorta was much smaller than the amount in the thoracic aorta. The media was excised from the descending thoracic aorta, followed by incubation with 1 mg/mL collagenase type II (Worthington Biochemical Corporation. Lakewood, NJ, USA) to remove endothelial and adventitial cells. The aortic medias were spread in medium containing 1 mg/mL collagenase type II, 0.5 mg/mL elastase type III (Sigma-Aldrich, St. Louis, MO, USA), and 20% fetal bovine serum (FBS). Cell suspensions were centrifuged at 500× *g* for 5 min, and the cell pellets were resuspended in Dulbecco’s modified eagle medium (DMEM) containing 100 U/mL penicillin, 100 μg/mL streptomycin, and 20% FBS. VSMCs were stimulated with 20 ng/mL PDGF-BB (PMG0045; Thermo Fisher Scientific) or 10 ng/mL TGF-β (766-MB; R&D Systems, Minneapolis, MN, USA), as shown in each experiment.

### 2.11. Western Blot Analysis

VSMCs were collected and lysed in an extraction buffer (50-mmol/L Tris-HCl (pH 7.5), 150-mmol/L NaCl, 50-mmol/L EDTA, 1% Triton X-100, and protease-phosphatase inhibitor mixture). The protein samples were subjected to sodium dodecylsulfate polyacrylamide gel electrophoresis (SDS-PAGE) and were then transferred to membranes that were subsequently incubated with primary antibodies against ERK (1:1000, 9102S; Cell Signaling Technology, Danvers, MA, USA), p-ERK (1:1000, 9101S; Cell Signaling Technology), JNK (1:1000, 9258S; Cell Signaling Technology), p-JNK (1:1000, 9255S; Cell Signaling Technology), Smad2 (1:1000, 3103S; Cell Signaling Technology), p-Smad2 (1:1000, 9255S; Cell Signaling Technology), PDGFRB (1:1000, 3169; Cell Signaling Technology), p-PDGFRB (1:1000, 3161; Cell Signaling Technology), Raf (1:1000, 9422; Cell Signaling Technology), p-Raf (1:1000, 9421; Cell Signaling Technology), MKP-1 (1:500, sc-373841; Santa Cruz Biotechnology), p-MKP-1 (1:500, 2857; Cell Signaling Technology), and β-actin (1:1000, T5168, B-5-1-2; Sigma-Aldrich). The immunoreactive proteins were envisioned using an enhanced chemiluminescence detection system (GE Healthcare Life Sciences, Marlborough, MA, USA), followed by exposure to a ChemiDoc MP Imaging System (Bio-Rad Laboratories). The band intensities were calculated using ImageJ software v1.50i (https://imagej.nih.gov/ij/index.html, accessed on 20 November 2021). β-Actin was used as a reference.

### 2.12. BrdU Assay

The BrdU assay was performed using the 5-bromodeoxyuridine (BrdU) ELISA Kit (ab126556, Abcam). Briefly, the cells were seeded in a 96-well plate at 5 × 10^3^ cells per well and were stimulated with 20 ng/mL PDGF (Thermo Fisher Scientific) for 24 h under starvation conditions. The cells were incubated with BrdU reagent for 6 h before the end of PDGF stimulation. After incubation with fixing solution, the anti-BrdU monoclonal antibody was added for 60 min at room temperature, followed by incubation with peroxidase goat anti-mouse IgG conjugate. The cells were then incubated with tetraliethylbenzidine peroxidase substrate for 30 min. After terminating the reaction by adding the stop solution, the absorbance was determined at 450 nm using a Tecan Infinite m200 plate reader.

### 2.13. Statistical Analysis

Data are expressed as the mean ± standard error of the mean (SEM). After examining the normality of distribution and equal variances, a Student’s *t*-test or analysis of variance (ANOVA) was used to analyze significant differences between the groups, followed by the Tukey–Kramer test. For the dependent variables: a non-aneurysmal portion vs. aneurysmal portion, PDGF stimulation, and phosphatase inhibitor treatment, significant differences were examined using two-way ANOVA. Statistical significance was set at *p* < 0.05. All analyses were performed using GraphPad Prism Ver 8.4.3 for Windows OS (GraphPad Software, LLC, San Diego CA, USA).

## 3. Results

### 3.1. Development of AAA Is Promoted in Defeated Mice

Eight-week-old male WT mice underwent RSD and were subjected to the behavior analysis test. The total immobility duration was markedly extended in the RSD-exposed mice than it was in the non-exposed mice (Appendix A). After the social interaction test, 28 of the 47 RSD-exposed mice showing an SIR less than 1.0 were certified as defeated mice. On the other hand, five non-exposed mice showing an SIR less than 1.0 were excluded in the following experiments (Appendix A). The body weight and hemodynamic parameters before CaCl_2_ application were equivalent between the two groups (Appendix A). The serum corticosterone concentrations were also similar between the two groups (Appendix A). After CaCl_2_ application, the maximum outer diameters of the AAA grew in a time-dependent manner in both the control and defeated mice. However, no discernible differences were detected between the two groups (Figure 1A,B). On the other hand, the circumferences of the external elastic membranes increased significantly in the defeated mice compared to those in the control mice 2 weeks after CaCl_2_ application (Figure 1C,D), suggesting that RSD promoted AAA expansion in the early phases of development.

### 3.2. Repeated Social Defeat Does Not Enhance the Acute Inflammatory Response after CaCl_2_ Application

We first examined the acute inflammatory response 1 week after CaCl_2_ application. Immunohistochemical staining for F4/80 and MMP-9 did not show any discernible differences between the two groups (Figure 2A,B). Consistently, the mRNA expression levels of the inflammatory cytokines were comparable between the two groups (Figure 2C). We further examined the MMP activity using ex vivo imaging 1 week after CaCl_2_ application; however, there were no discernable differences between the two groups (Figure 2D,E), suggesting that the acute inflammatory response after CaCl_2_ application was equivalent between the two groups and that it was not likely to contribute to augmented AAA expansion in defeated mice.

### 3.3. Repeated Social Defeat Inhibits Perivascular Fibrotic Healing after the Acute Inflammatory Response

We focused on fibrotic wound healing after the acute inflammatory response to clarify the augmented AAA expansion mechanism in defeated mice. At 1 and 4 weeks after CaCl_2_ application, there no significant differences were observed in the fibrotic areas between the two groups (Figure 3A,B). In contrast, the fibrotic area in the aneurysmal portion was significantly reduced in the defeated mice compared to in the control mice 2 weeks after CaCl_2_ application, whereas no significant differences were observed between the two groups in the non-aneurysmal portion (Figure 3C,D). We further examined the α-SMA-positive areas in the tunica media and adventitia (Figure 3E). In the tunica media, the α-SMA-positive areas in the aneurysmal lesions were scarcely observed in both the control and defeated mice (Figure 3F). However, in the adventitia, the α-SMA-positive areas in aneurysmal lesions were markedly decreased in the defeated mice compared to in the control mice, whereas there were no discernable differences observed in the non-aneurysmal portions between the two groups (Figure 3G). These findings suggest that fibrotic wound healing was inhibited in the defeated mice, independent of the reduction in the α-SMA-positive areas in the tunica media.

### 3.4. PDGF-Induced ERK Phosphorylation and BrdU Incorporation in VSMCs Are Inhibited in Defeated Mice

To examine the mechanisms of impaired fibrotic healing in defeated mice, we cultured VSMCs from the thoracic aorta of control and defeated mice. There were no discernible differences in proliferation under the conventional culture conditions. However, the ERK phosphorylation after PDGF stimulation was significantly lower in the VSMCs of defeated mice than it was in the control mice (Figure 4A,B). Consistently, PDGF-induced BrdU uptake in the VSMCs of defeated mice was significantly lower than it was in control mice (Figure 4C). We examined the phosphorylation of JNK-1 after PDGF stimulation; however, no differences were observed between the two groups (Appendix A). Furthermore, Smad-2 phosphorylation after TGF-β stimulation was comparable between the two groups (Appendix A). These findings suggest that the impaired VSMC proliferation in the defeated mice is implicated in the PDGF-activated ERK signaling pathway.

### 3.5. PDGF-Induced MKP-1 Expression in VSMCs Is Augmented in Defeated Mice

To clarify the molecular mechanisms involved in the decreased ERK phosphorylation after PDGF stimulation, we examined the upstream ERK phosphorylation pathways. However, the PDGFRB and Raf phosphorylation were comparable between the two groups (Appendix A). We next examined the protein expression levels of MKP-1, which is a negative regulator of MAPK. We first examined the baseline MKP-1 protein expression levels in VSMC without PDGF stimulation; however, MKP-1 did not show high levels of expression in either of the VSMC groups (Appendix A). MKP-1 expression after PDGF stimulation was significantly higher in the VSMCs of defeated mice than it was in the control mice (Figure 5A,B). The MKP-1protein expression level was dependent on the stability that it gained via ERK-mediated phosphorylation [26]. Consistently, the PDGF-induced MKP-1 phosphorylation was markedly increased in the VSMCs of the defeated mice compared to in the control mice (Figure 5C,D), whereas the PDGF-induced mRNA expression level of MKP-1 was equivalent between the two groups (Figure 5E). Finally, we examined the effect of the phosphatase inhibitor BCI on impaired ERK phosphorylation in the VSMCs of the defeated mice. After treatment with BCI, the PDGF-induced ERK phosphorylation was comparable between the two groups (Figure 5F,G). These findings further suggest that the PDGF-induced ERK phosphorylation in the VSMCs of the defeated mice is eliminated by the augmented MKP-1 expression.

### 3.6. MKP-1 Expression in AAA Is Exaggerated in Defeated Mice

We examined MKP-1 expression in AAA and found that MKP-1 mRNA expression was significantly higher in defeated mice than in control mice (Figure 6A). Consistently, the percentage of MKP-1-positive nuclei was markedly higher in the defeated mice than in the control mice (Figure 6B,C). Furthermore, the double staining of MKP-1 and α-SMA revealed that the percentage of the MKP-1-positive stained nuclei in the α-SMA-positive cells was markedly higher in the defeated mice than in the control mice (Figure 6D,E). These findings indicate that the impaired accumulation of α-SMA-positive cells and reduced fibrotic response in the defeated mice was mediated by augmented MKP-1 expression.

### 3.7. Treatment with MKP-1 Inhibitor Attenuated Exaggerated AAA Development in Defeated Mice

To examine the effect of MKP-1 on the exaggerated aneurysmal expansion in defeated mice, an MKP-1 inhibitor, BCI, was administered after CaCl_2_ application. No significant differences were observed in neither the maximum outer diameters nor the circumferences of the external elastic membrane between the two groups of BCI-treated mice (Figure 7A–D). The perivascular fibrotic area was also comparable between the two groups (Figure 7E,F). These findings support the notion that augmented MKP-1 expression contributes to the attenuated early fibrotic response along with the exaggerated expansion of AAA in defeated mice.

## 4. Discussion

We showed that RSD enhances CaCl_2_-induced AAA development and impaired fibrotic wound healing following an acute inflammatory response. Consistently, the PDGF-induced ERK phosphorylation and subsequent DNA synthesis in the primarily cultured VSMCs from the defeated mice was significantly reduced compared to in the control mice. The PDGF-induced MKP-1 protein expression was noticeably higher in the VSMCs from the defeated mice than in it was in the control mice, and treatment with the phosphatase inhibitor completely restored ERK phosphorylation to the same extent as seen in the control mice. Finally, the mRNA and protein expression of MKP-1 in AAA was markedly enhanced by RSD. These findings suggest that RSD exaggerates AAA development by eliminating fibrotic wound healing via the MAPK–MKP-1 signaling pathway, indicating that the restoration of the wound healing process could be a potential therapeutic target in depression-related AAA expansion.

Maegdefessel et al. demonstrated that an early fibrotic response in the abdominal aortic wall plays a crucial role in inhibiting AAA progression in gene-manipulated mice [27]. The aortic expression of miR-29b, a known target for collagen-synthesis genes, was significantly decreased in both the porcine pancreatic elastase infusion model and in the Angiotensin II infusion model along with AAA development. Treatment with anti-miR-29b treatment significantly augmented collagen deposition, leading to a reduction in AAA development, and vice versa, as well as an overexpression of miR-29b. Lindeman et al. also demonstrated that the adventitial collagen fibers that encage the vessel prevent the vessel from overstretching and that their functional failure is associated with the expansion of the abdominal aneurysm in patients with AAA [28]. These findings suggest that adventitial fibrosis following acute inflammation plays an important role in preventing the progression of AAA in humans as well as in animal models.

Cole-King et al. were the first to report that the severity of depression was significantly correlated with delayed wound healing after punch biopsy [29]. Numerous human studies have shown that the patients who experience the highest levels of depression and anxiety are more likely to exhibit impaired wound healing after surgery [30,31]. In animal studies using various types of wound healing models, psychological stress also delayed the wound healing process, implicating the inflammatory cytokines and hypothalamic–pituitary–adrenal (HPA) activity [30]. Fibrosis is a multifactorial process, and macrophage-mediated inflammatory responses contribute to subsequent inflammation-associated fibrosis [32]. Indeed, Goldberg et al. showed that inflammatory cytokines such as TNF-α can inhibit myofibroblast differentiation and can contribute to a delay in the wound healing process [33]. However, given that the mRNA expression of inflammatory cytokines and the serum concentration of corticosterone were not affected by RSD, it is likely that the impaired wound healing observed in defeated mice is independent of the acute inflammatory response.

Duric et al. showed a significant increase in the expression of MKP-1, a negative regulator of ERK phosphorylation, in the hippocampal subfields of postmortem tissue from subjects with major depressive disorder (MDD) [34]. They also demonstrated that the hippocampal expression of MKP-1 was increased in rat and mouse models of depression. Likewise, Wang et al. showed that ERK1/2 activity was significantly downregulated in the prefrontal cortex and hippocampus and that it was accompanied by enhanced MKP activity in patients with depression and in animal models of depression [35]. These findings suggest that the MAPK–MKP-1 signaling pathway in the central nervous system (CNS) plays a crucial role in the pathogenesis of depression. The MAPK–MKP-1 pathway plays a crucial role in VSMC proliferation and migration during vascular remodeling, including in the fibrotic response after arterial injury [36,37]. The phenotypic modulation of VSMCs via the MAPK–MKP-1 signaling pathway is likely to be responsible for the impaired fibrotic response in defeated mice, which was also observed in the neural cells in the CNS of patients with depression.

Because the genes and proteins comprising MKP-1 have a very short half-life, MKP-1 activity is largely dependent on protein stability [38,39]. MKP-1 stability and activity are augmented by the direct phosphorylation of serine 359 and serine 364 through their interaction with ERK1/2, which exerts a well-defined negative-feedback MAPK control mechanism [38]. MKP-1 was highly expressed in the VSMCs of arteries in vivo [40]. The proliferation and migration of the VSMCs following vascular injury may be attributed, at least in part, to a decrease in MKP-1 expression [41]. Based on our findings that treatment with a phosphatase inhibitor completely restored the ERK phosphorylation that had been in the VSMCs from defeated mice and that it augmented MKP-1 gene and protein expression in the perivascular area after CaCl_2_ application in the defeated mice, it is likely that the RSD-induced modulation of the MAPK–MKP-1 pathway in VSMCs is implicated in the impaired migration and replication of α-SMA-positive cells, thereby contributing to the decreased fibrotic wound healing process and subsequent AAA expansion in defeated mice.

Epigenetic changes that are modulated by microRNAs (miRNAs) have been intensively explored in patients with major depressive disorder (MDD) and in patients with AAA. Maegdefessel et al. showed that miR29-b is profoundly implicated in AAA expansion in human and animal models [27], and Wan et al. examined differential miRNA expression in the cerebrospinal fluid (CSF) of patients with MDD [42]. They showed that the miR-29b-3p expression levels in the CSF were significantly higher in depressed patients than they were in the control subjects, suggesting the involvement of increased miR-29b expression in depression-related AAA development. Compared to miRNA-29b, the miRNA-146a expression level in patients with MMD has been studied more extensively [43,44]. Hung et al. showed that the miRNA-146a levels in the peripheral blood monocytes from MDD patients were significantly lower than they were in healthy controls, which was partially restored after antidepressant treatment [45]. They focused on the role of miRNA-146b in the inflammatory response via the TLR-4 pathway that promotes the excessive production of pro-inflammatory cytokines. In contrast, Lopez et al. showed that miRNA-146a expression was upregulated in brain tissue from MDD patients who had committed suicide [46]. They also observed a significant downregulation in miR-146a-5p expression in blood samples from depressed patients who responded to antidepressant treatment. Furthermore, they revealed the expression levels of the miR-146a-5p-regulated genes involved in the MAPK pathway. Given that miRNA-146a expression was upregulated in AAA tissue samples from patients [47], augmented miRNA-146a expression might lead to the dysregulation of the MAPK–MKP-1 signaling pathway that impairs the fibrotic response as well as the depression-like behavior observed in defeated mice.

ERK1/2 activation has been shown to promote the development of aortic aneurysm in various experimental animal models. Holm et al. [48] reported that TGF-β-induced ERK1/2 activation promoted aortic root dilatation in a mouse model of Marfan syndrome. Ghosh et al. [49] also demonstrated that ERK1/2 activation exaggerated aneurysmal dilatation in a murine elastase infusion model by enhancing MMP activity. Recently, Peng et al. [50] reported that ERK1/2 activation is involved in the VSMC phenotype switching from a contractile to a synthetic phenotype, leading to an increase in proliferation, migration, and MMP activity. However, several kinds of inflammatory cytokines and growth factors are involved in ERK1/2 activation through different receptors and subsequent signal transduction pathways. The intensity and duration of ERK1/2 activation are ligand-specific and are able to determine cellular responses in a cell-specific manner [51]. From this point of view, ERK1/2 activation that is elicited by noncanonical TGF-β signaling or proinflammatory cytokines is preferentially involved in MMP secretions, thereby promoting aortic aneurysm in conventional experimental animal models. In contrast, our findings suggest that PDGF-stimulated ERK1/2 activation is specifically implicated in reducing depression-related AAA development.

The present study has methodological implications. First, we retrieved the primary cultured VSMCs from murine thoracic aortic tissues, but not from the infrarenal aorta. Phenotypic variations between the VSMCs from the thoracic aorta and abdominal aorta have long been reported in terms of embryological origins and subsequent transcriptomic profiles. However, there was less tissue in the infrarenal aorta than there was in the thoracic aorta. Therefore, we examined the primary cultured VSMCs from the descending thoracic aorta, the characterization of which has been well established. Second, adventitial fibroblasts play a crucial role in vascular remodeling [52]. We first focused on the adventitial fibroblasts in the descending thoracic aorta. However, the flow cytometric analysis and the sorting of the adventitial fibroblasts were technically difficult due to the small amount of tissue that could be collected from the descending thoracic aorta. We therefore examined the VSMCs instead of the adventitial fibroblasts. Furthermore, because MKP-1 is a negative regulator of JNK and p-38 as well as of ERK, we cannot exclude the possibility that the augmented MKP-1 expression affected JNK, p-38, ERK phosphorylation in the in vivo experiment. Finally, further phenotypic characterizations of the VSMCs and adventitial fibroblasts, including their migration response and collagen deposition, as well as the involvement of canonical TGF-β signaling need to be investigated in future studies using Cre-driver defeated mice.

We focused on the link between psychological stress and AAA development in the context of wound healing after an inflammatory response. We showed that psychological stress could promote AAA expansion, at least in part, by eliminating perivascular fibrosis through the MAPK–MKP-1 signaling pathway, as is the case for neuronal activity in the brain tissue of patients with MDD. Our findings provide novel insight into the mechanisms of depression-related AAA development, showing that the MAPK–MKP-1 pathway that is involved in the wound healing process can be a potential therapeutic target in depression-related AAA.

## Figures and Tables

**Figure 1 cells-11-00732-f001:**
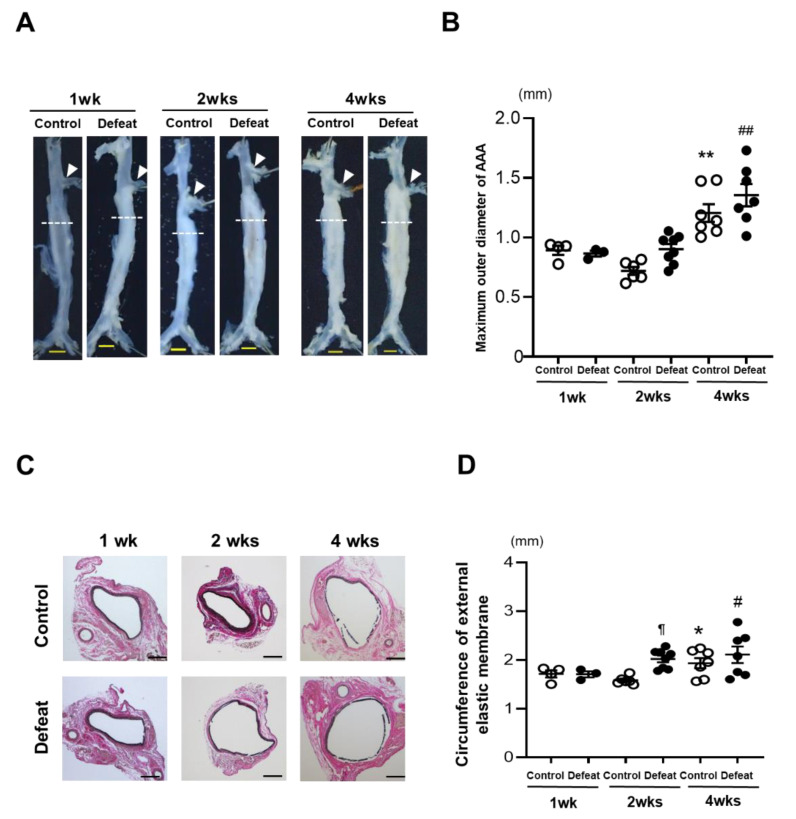
Repeated social defeat promotes the expansion of CaCl_2_-induced AAA. (**A**,**B**) Representative photographs and quantitative analysis of the maximum outer diameters at 1, 2, and 4 weeks after CaCl_2_ application. Arrowheads indicate the orifice of the left renal artery. A dotted line shows the level of the maximum outer diameter. Scale bar = 1 mm. Values represent the mean ± SEM for four control and three defeated mice at 1 week, six control and for eight defeated mice at 2 weeks, and seven control and seven defeated mice at 4 weeks. ** *p* < 0.01 vs. control at 1 and 2 weeks after CaCl_2_ application. ^##^
*p* < 0.01 vs. defeat at 1 and 2 weeks after CaCl_2_ application; two-way repeated measures ANOVA with the Tukey–Kramer post hoc test. (**C**,**D**) Representative photographs of Elastica van Gieson stain and quantitative analysis of the circumferences of the external elastic membranes at 1, 2, and 4 weeks after CaCl_2_ application. Scale bar = 200 μm. Values represent the mean ± SEM for four control and three defeated mice at 1 week, six control and eight defeated mice at 2 weeks, and seven control and seven defeated mice at 4 weeks. * *p* < 0.05 vs. control at 1 week after CaCl_2_ application. ^#^
*p* < 0.05 vs. defeat at 1 week after CaCl_2_ application. ^¶^
*p* < 0.05 vs. control at 2 weeks after CaCl_2_ application; two-way repeated measures ANOVA with the Tukey–Kramer post hoc test.

**Figure 2 cells-11-00732-f002:**
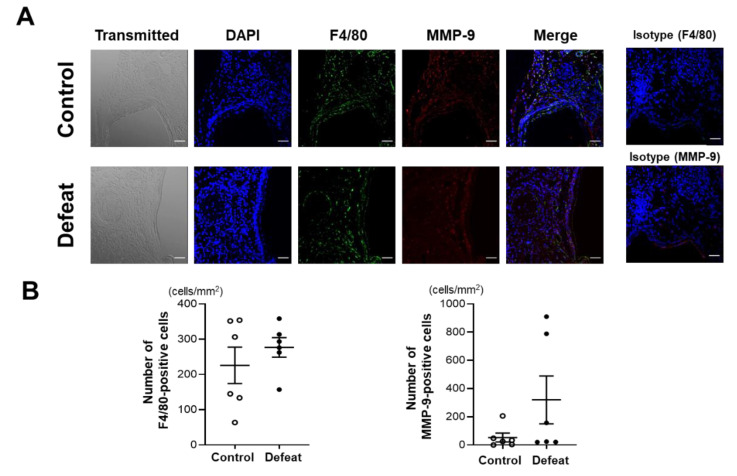
Repeated social defeat does not enhance the acute inflammatory response after CaCl_2_ application. (**A**,**B**) Representative fluorescent images of F4/80-positive and MMP-9-positive cells and a quantitative analysis in AAA from control and defeated mice at 1 week after CaCl_2_ application. Scale bar = 25 μm. Values represent the mean ± SEM for six control and six defeated mice. (**C**) Quantitative PCR analysis of mRNA expression levels of F4/80, TNF-α, IL-1β, IL-4, and TGF-β in AAA at 1 week after CaCl_2_ application. Values represent the mean ± SEM for four control and three defeated mice. * *p* < 0.05 vs. control; Student’s *t*-test. (**D**,**E**) Representative ex vivo images of AAA and quantitative analysis of MMP activity at 1 week after CaCl_2_ application. Arrows show the orifice of the left renal artery. Values represent the mean ± SEM for four control and three defeated mice.

**Figure 3 cells-11-00732-f003:**
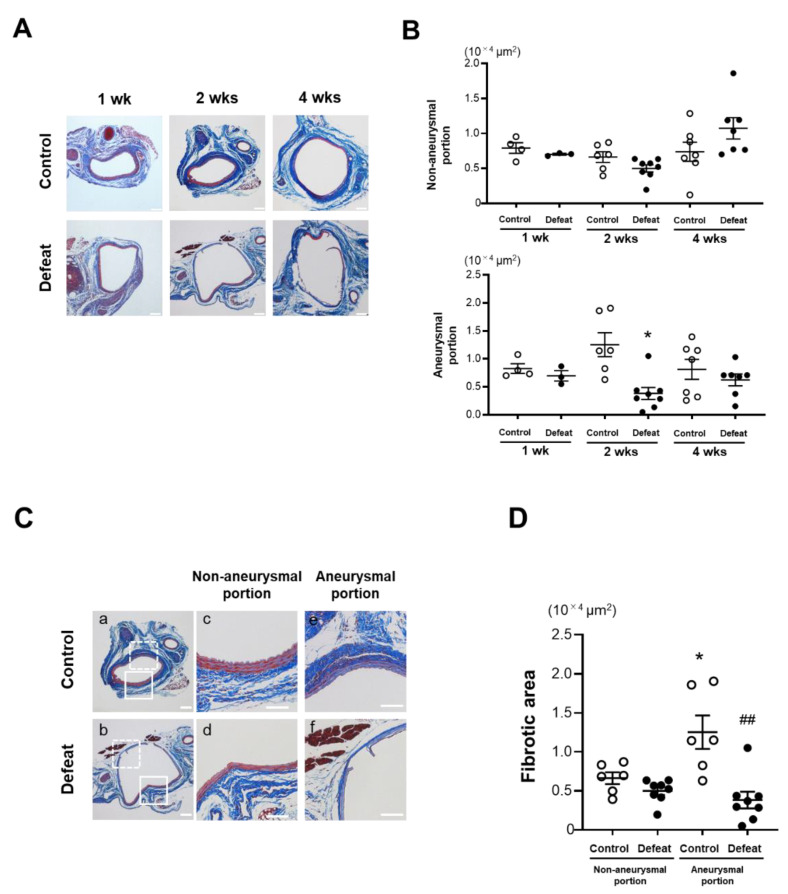
Repeated social defeat inhibits perivascular fibrotic healing after the acute inflammatory response. (**A**,**B**) Representative photographs of Masson’s trichrome stain and quantitative analysis of fibrotic areas in control and defeated mice 1, 2, and 4 weeks after CaCl_2_ application. Scale bar = 200 μm. Values are mean ± SEM for four control and three defeated mice at 1 week, six control and eight defeated mice at 2 weeks, and seven control and seven defeated mice at 4 weeks. * *p* < 0.05 vs. control at 2 weeks after CaCl_2_ application; two-way repeated measures ANOVA with the Tukey–Kramer post hoc test. (**C**,**D**) Representative photographs of Masson’s trichrome stain and quantitative analysis of fibrotic areas in control and defeated mice 2 weeks after CaCl_2_ application. High magnification images (**c**,**d**) showing the area surrounded by a solid line in panel (**a**,**b**), respectively. High magnification images (**e**,**f**) showing the area surrounded by a broken line in panels a and b, respectively. Scale bar = 200 μm. Values are mean ± SEM for six control and seven defeated mice. * *p* < 0.05 vs. control in non-aneurysmal portion. ^##^
*p* < 0.01 vs. control in aneurysmal portion; two-way ANOVA with the Tukey–Kramer post hoc test. (**E**–**G**) Representative fluorescent images of α-SMA and quantitative analysis of α-SMA -positive areas in control and defeated mice at 2 weeks after CaCl_2_ application. L: lumen, M: media, A: adventitia. A broken line indicates an external elastic plate. Scale bar = 25 μm. Values are mean ± SEM for five control and six defeated mice. ** *p* < 0.01 vs. control in non-aneurysmal portion. ^#^
*p* < 0.05, ^##^
*p* < 0.01 vs. control in aneurysmal portion; two-way ANOVA with the Tukey–Kramer post hoc test.

**Figure 4 cells-11-00732-f004:**
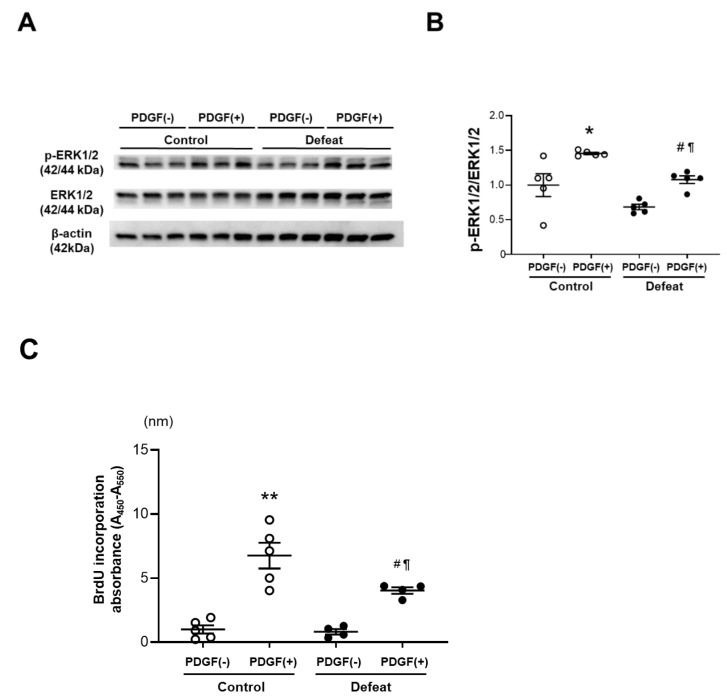
Repeated social defeat inhibits PDGF-induced ERK phosphorylation and BrdU incorporation in primary cultured VSMCs. (**A**,**B**) Representative Western blot of ERK phosphorylation in VSMCs at 30 min after PDGF stimulation and a quantitative analysis of protein expression. Values are the mean ± SEM for five control and five defeated mice. * *p* < 0.05 vs. control without PDGF stimulation. ^#^
*p* < 0.05 vs. defeat without PDGF stimulation. ^¶^
*p* < 0.05 vs. control with PDGF stimulation; two-way repeated measures ANOVA with the Tukey–Kramer post hoc test. (**C**) BrdU incorporation after PDGF stimulation. Values are the mean ± SEM for five control and four defeated mice. ** *p* < 0.01 vs. control without PDGF stimulation. ^#^
*p* < 0.05 vs. defeat without PDGF stimulation. ^¶^
*p* < 0.05 vs. control with PDGF stimulation; two-way repeated measures ANOVA with the Tukey–Kramer post hoc test.

**Figure 5 cells-11-00732-f005:**
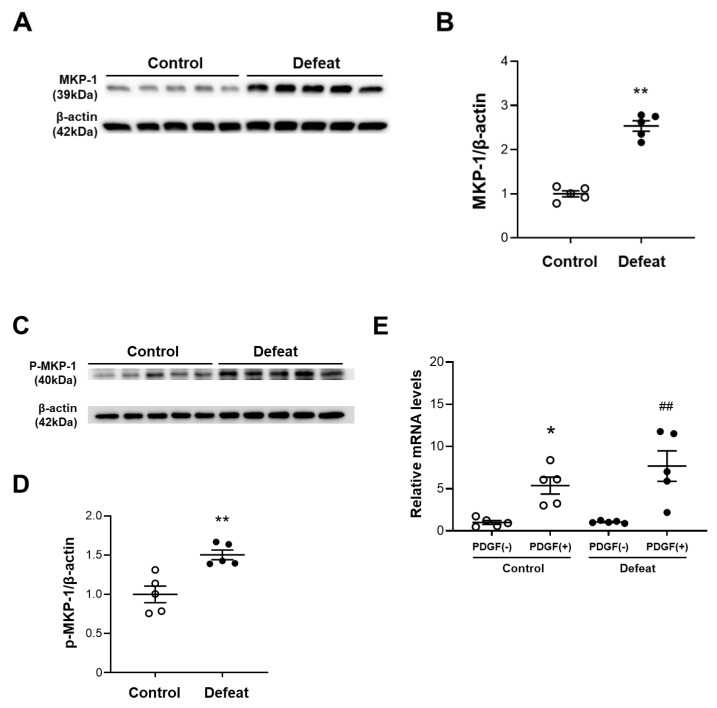
Repeated social defeat augments MKP-1 protein expression levels in VSMCs by enhancing MKP-1 stabilization. (**A**,**B**) Representative Western blot of MKP-1 in VSMCs at 30 min after PDGF stimulation and a quantitative analysis of protein expression. Values are the mean ± SEM for five control and five defeated mice. ** *p* < 0.01 vs. control; Student’s *t*-test. (**C**,**D**) Representative Western blot of the MKP-1 phosphorylation in the VSMCs at 30 min after PDGF stimulation and a quantitative analysis of protein expression. Values are the mean ± SEM for five control and five defeated mice. ** *p* < 0.01 vs. control; Student’s *t*-test. (**E**) Quantitative PCR analysis of mRNA expression level of MKP-1 in the VSMCs. Values represent the mean ± SEM for five control and five defeated mice. * *p* < 0.05 vs. control without PDGF stimulation. ^##^
*p* < 0.01 vs. defeat without PDGF stimulation; two-way repeated measures of ANOVA with the Tukey–Kramer post hoc test. (**F**,**G**) Representative Western blot of PDGF-induced ERK phosphorylation in VSMCs with or without BCI treatment and a quantitative analysis of protein expression. Values are the mean ± SEM for three control and three defeated mice in each group. * *p* < 0.05 vs. control without BCI treatment. ^#^
*p* < 0.05 vs. defeat without BCI treatment; two-way repeated measures of ANOVA with the Tukey–Kramer post hoc test. BCI, dual specificity MAPK phosphatase (DUSP-MKP) inhibitor.

**Figure 6 cells-11-00732-f006:**
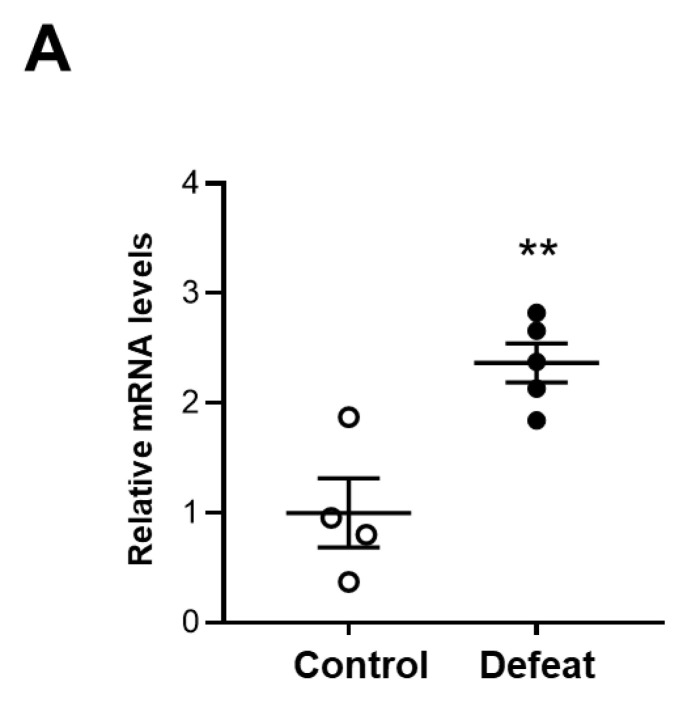
Repeated social defeat increases MKP-1 gene and protein expression levels in AAA. (**A**) Quantitative PCR analysis of mRNA expression levels of MKP-1 in AAA 2 weeks after CaCl_2_ application. Values represent the mean ± SEM for four control and five defeated mice. ** *p* < 0.01 vs. control; Student’s *t*-test. (**B**,**C**) Representative fluorescent images of MKP-1-positive cells and a quantitative analysis of AAA from control and defeated mice 2 weeks after CaCl_2_ application. Scale bar = 5 μm. Arrow heads indicate MKP-1-positive cells. Values represent the mean ± SEM for six control and eight defeated mice. ** *p* < 0.01 vs. control; Student’s *t*-test. (**D**,**E**) Representative fluorescent images of MKP-1-positive and α-SMA-positive cells as well as a quantitative analysis of AAA from control and defeated mice 2 weeks after CaCl_2_ application. Scale bar = 5 μm. Arrow heads indicate MKP-1-positive cells. Values represent the mean ± SEM for four control and four defeated mice. ** *p* < 0.01 vs. control; Student’s *t*-test.

**Figure 7 cells-11-00732-f007:**
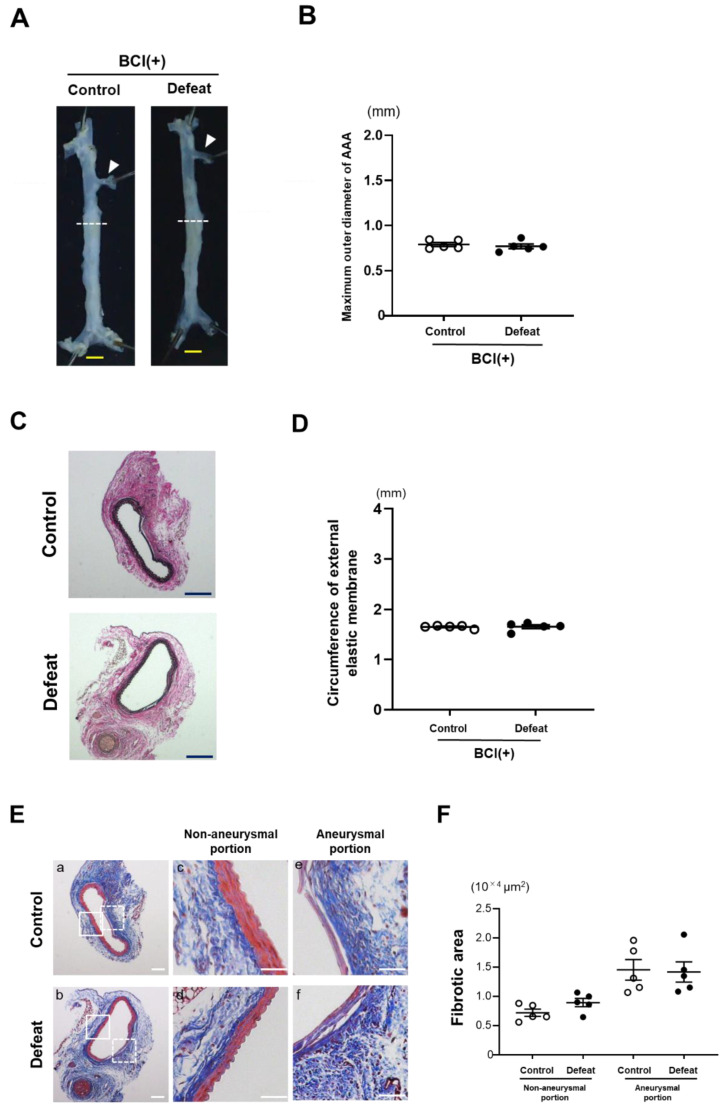
Augmented aneurysm expansion in defeated mice is eliminated after treatment with MKP-1 inhibitor. (**A**,**B**) Representative photographs and quantitative analysis of the maximum outer diameters 2 weeks after CaCl_2_ application. Arrowheads indicate the orifice of the left renal artery. A dotted line shows the level of the maximum outer diameter. Scale bar = 1 mm. Values represent the mean ± SEM for five control and five defeated mice. (**C**,**D**) Representative photographs of Elastica van Gieson stain and quantitative analysis of the circumferences of the external elastic membrane 2 weeks after CaCl_2_ application. Scale bar = 200 μm. Values represent the mean ± SEM for five control and five defeated mice. (**E**,**F**) Representative photographs of Masson’s trichrome stain and quantitative analysis of fibrotic areas in control and defeated mice 2 weeks after CaCl_2_ application. High magnification images (**c**,**d**) showing the area surrounded by a solid line in panel (**a**,**b**), respectively. High magnification images (**e**,**f**) showing the area surrounded by a broken line in panel (**a**,**b**), respectively. Scale bar = 100 μm. Values are mean ± SEM for five control and five defeated mice.

## Data Availability

Not applicable.

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
