# Peer review of "Repeated Social Defeat Enhances CaCl2-Induced Abdominal Aortic Aneurysm Expansion by Inhibiting the Early Fibrotic Response via the MAPK-MKP-1 Pathway"

_cells, 2022, doi:10.3390/cells11040732_

Round 1
Reviewer 1 Report
In this manuscript, the authors evaluated the role of repeated social defeat on AAA development. Using the CaCl2-induced AAA model, they showed RSD-caused increase in aortic circumference, explained by lowered periaortic fibrosis caused by MKP-1-driven attenuation of ERK phosphorylation. While the investigation of the mechanisms underlying depression-related AAA development is interesting and potentially clinically relevant, the study has several aspects that need to be addressed. My specific comments are summarized below:
- I did not fully understand the description of the treatment for the control mice. Were all the control mice housed together, or were they housed separately? Did they change cages throughout the experiment? Did they have any interaction with aggressor mice?
- How were the aneurysmal and non-aneurysmal portions of the aortic sections defined? Was any kind of standardization used?
- The SMA staining presented in Fig. 3E is not of high quality. Were the tissues in the representative pictures oriented the same way (i.e. lumen side up or lumen size down)? Please make sure that is the case and mark the media and adventitia on the figures.
- To perform cell experiments, the authors isolated VSMCs from the thoracic aorta. Considering the differences between thoracic and abdominal aortic SMCs, why did the authors decide not to use suprarenal aortic tissue instead, or infrarenal aortas from a separate group of mice?
- Why did the authors focus on VSMCs in studying the mechanisms of RSD-mediated fibrosis? In AAA, VSMCs are not the only source of ECM-secreting myofibroblasts, and other cell types, particularly adventitial fibroblasts, are known as abundant producers of collagen and other fibrotic components. Especially considering that the reported differences in SMA-positive cells were found in the adventitia, it is essential that the authors evaluate and compare the potential contribution of aortic fibroblasts vs VSMCs on the observed fibrotic phenotype.
- It seems like in Fig. 4B, ERK phosphorylation is decreased (without reaching significance) also in the PDGF(-) defeat group. Thus, the difference in pERK between the VSMCs with or without PDGF stimulation seems to be conserved between control and defeat groups. Can the authors comment on that? Also, the change in pERK seems quite subtle – were kinase pathways other than ERK and JNK (PI3K, p38) affected by RSD? Furthermore, what is the specificity of MKP-1; is ERK the only target, or could other MAPK pathways have been affected?
- The authors report no changes in SMAD2 activation, but as SMAD2 and SMAD3 are known to have overlapping but also distinct functions, SMAD3 activation should be evaluated as well. Was pSMAD2/pSMAD3 levels affected in AAA tissues at the relevant time point?
- What are the MKP-1 baseline levels in naïve VSMC without PDGF stimulation?
- What was the effect of MKP-1 on VSMC migration and collagen expression/deposition?
- Finally, MKP-1 inhibition should be used in vivo to confirm the VSMC in vitro results.
Author Response
Please find the attached file below.

Reviewer 2 Report
The study by Kubota et al sets out to investigate the effect of depression on development of abdominal aortic aneurysm (AAA), and to delineate any associated molecular mechanism. An in vivo mouse model was used, whereby mice were subjected to repeated social defeat (RSD) to induce depressive-like behaviour, followed by calcium chloride induction of AAA (which is one of several standard rodent models of AAA). Histological and immunohistochemical analyses were then undertaken to see effects on AAA development, fibrosis, metalloproteinase activity; and cells were then prepared from aortic media and stimulated with PDGF and TGF-beta to look for differences in signalling molecules. The authors present observations showing an increase in AAA in RSD mice due to reduction in fibrosis that may be due to in part, reduction in p42/44 MAPK activation by the MAPK phosphatase MKP-1.
Major criticism:
Whilst the present study suggests the involvement of the p42/44 MAPK/MKP-1 pathway in reducing AAA, this observation contradicts other previously published reports (e.g. J Am Coll Surg. 2012 Nov, 215(5): 668-690.e1; and Science 2011 Apr 15: 332(6027); 358-361) that infer a role for these kinases in promoting aneurysms. I feel that the authors ought to address some of these reports in their discussion.
There are also some minor points which the authors need to address:
- The authors should make it clear in the figures of the Western Blots what they mean by “relative intensity”- i.e. relative to B-actin?
- Overall, the manuscript is clear and well written, but may still benefit from editing especially in the introduction
Author Response
Response to Reviewer #2
The study by Kubota et al sets out to investigate the effect of depression on development of abdominal aortic aneurysm (AAA), and to delineate any associated molecular mechanism. An in vivo mouse model was used, whereby mice were subjected to repeated social defeat (RSD) to induce depressive-like behaviour, followed by calcium chloride induction of AAA (which is one of several standard rodent models of AAA). Histological and immunohistochemical analyses were then undertaken to see effects on AAA development, fibrosis, metalloproteinase activity; and cells were then prepared from aortic media and stimulated with PDGF and TGF-beta to look for differences in signalling molecules. The authors present observations showing an increase in AAA in RSD mice due to reduction in fibrosis that may be due to in part, reduction in p42/44 MAPK activation by the MAPK phosphatase MKP-1.
Major criticism:
Whilst the present study suggests the involvement of the p42/44 MAPK/MKP-1 pathway in reducing AAA, this observation contradicts other previously published reports (e.g. J Am Coll Surg. 2012 Nov, 215(5): 668-690.e1; and Science 2011 Apr 15: 332(6027); 358-361) that infer a role for these kinases in promoting aneurysms. I feel that the authors ought to address some of these reports in their discussion.
Response:
According to the Reviewer’s important comment, we addressed the role of ERK1/2 activation in aortic aneurysm development in the previously reported papers, and discussed the incongruity with our results.
ERK1/2 activation has been shown to promote the development of aortic aneurysm in various experimental animal models. Holm et al. [Ref. 48 in the text, see below] reported that TGF-β-induced ERK1/2 activation promote aortic root dilatation in a mouse model of Marfan syndrome. Ghosh et al. [Ref. 49 in the text, see below] also demonstrated that ERK1/2 activation exaggerated aneurysmal dilatation in murine elastase infusion model by enhancing MMPs activity. Recently, Peng et al. [Ref. 50 in the text, see below] reported that ERK1/2 activation is involved in the VSMCs phenotype switching from a contractile to a synthetic phenotype, leading to an increase in proliferation, migration, and MMPs activity. However, several kinds of inflammatory cytokines and growth factors are involved in ERK1/2 activation through different receptors and subsequent signal transduction pathways. The intensity and duration of ERK1/2 activation are dependent on a ligand-specific manner and determine the cellular responses in a cell-specific manner [Ref. 51 in the text, see below]. From this point of view, ERK1/2 activations elicited by noncanonical TGF-β signaling or proinflammatory cytokines are preferentially involved in MMPs secretions thereby promoting aortic aneurysm in conventional experimental animal models. In contrast, our findings suggest that PDGF-stimulated ERK1/2 activation is specifically implicated in the pathogenesis of depression-related AAA development.
Reference
- Holm, T.M.; Habashi, J.P.; Doyle, J.J.; Bedja, D.; Chen, Y.; van Erp, C.; Lindsay, M.E.; Kim, D.; Schoenhoff, F.; Cohn, R.D.; et al. Noncanonical TGFβ signaling contributes to aortic aneurysm progression in Marfan syndrome mice. Science. 2011, 332, 358-361. doi: 10.1126/science.1192149.
- Ghosh, A.; DiMusto, P.D.; Ehrlichman, L.K.; Sadiq, O.; McEvoy, B.; Futchko, J.S.; Henke, P.K.; Eliason, J.L.; Upchurch, G.R. Jr. The role of extracellular signal-related kinase during abdominal aortic aneurysm formation. Am. Coll. Surg. 2012, 215, 668-680.e1. doi: 10.1016/j.jamcollsurg.2012.06.414.
- Peng, H.; Zhang, K.; Liu, Z.; Xu, Q.; You, B.; Li, C.; Cao, J.; Zhou, H.; Li, X.; Chen, J.; et al. VPO1 Modulates Vascular Smooth Muscle Cell Phenotypic Switch by Activating Extracellular Signal-regulated Kinase 1/2 (ERK 1/2) in Abdominal Aortic Aneurysms. Am. Heart Assoc. 2018, 7:e010069. doi: 10.1161/JAHA.118.010069.
- Marshall, C.J. Specificity of receptor tyrosine kinase signaling: transient versus sustained extracellular signal-regulated kinase activation. Cell. 1995, 80, 179-185. doi: 10.1016/0092-8674(95)90401-8.
These were stated in the line 574 in page 22 in the Revised Discussion section (Track changes version).
Minor points:
The authors should make it clear in the figures of the Western Blots what they mean by “relative intensity”- i.e. relative to B-actin?
Response:
According to the Reviewer’s comment, we changed the term “relative intensity” to an accurate label.
These were shown in the Revised Figures 4 and 5 and Supplementary Figures 4, 5, and 6.
All authors are very grateful to Reviewer #2 for carefully reviewed comments and suggestions. We strongly hope that this revised version will be acceptable by the Reviewer #2.
Round 2
Reviewer 1 Report
While I appreciate the changes made by the authors, especially new (though obviously previously generated) experimental data incorporated as Fig. 7, the overall link between MKP-1 upregulation in RSD mice and ERK activation has not been proven and remains circumstantial. Therefore, it needs to be studied in more detail, and the potential contributions of other biological factors and processes must be taken into account.
My detailed response to authors' rebuttal is summarized below:
Comments 4: should be addressed in the Discussion section, potentially as the limitation of the study.
Comment 5: there are established protocols for adventitial fibroblast isolation and culture. The authors should discuss the choice of the studied cell type and adress the potential contribution of adventitial fibroblasts to the observed effects in the Discussion section, potentially as the limitation of the study.
Comment 6: the authors suggest that PDGF-induced increased MKP-1 activation is mediated by ERK but have not proven that. To do that, the impact of ERK inhibitors on MKP-1 phosphorylation would need to be assessed.
As I stated before, small differences in pERK/ERK ratios suggest that other pathways might be involved, especially that the authors themselves admit that MKP-1 is not ERK-specific but can affect other MAP kinases as well. Therefore, the potential activation of other kinase pathways such as PI3K and p38 needs to be investigated.
Comment 7: The authors’ response was not satisfactory. Given the crucial role of canonical TGF-β signaling in fibrotic response, also within the context of AAA, the potential SMAD3 involvement should be evaluated, preferably both in vitro and in vivo.
Comment 8: The baseline MKP-1 protein levels should be shown as the figure and/or described in the Results section.
Comment 9: The authors’ response is not satisfactory. The effects of MKP-1 on VSMC migration are important to evaluate, especially considering the reported adventitial localization of SMA-positive cells in vivo. Moreover, to strengthen the presented proliferation angle, in situ proliferation (BrdU or otherwise) should be assessed on AAA tissues from control and RSD animals.
Similarly, MKP-1 influence on VSMC collagen expression/deposition should be evaluated to investigate the link between studied VSMCs and the main endpoint of the in vivo study (attenuated fibrotic response).
Author Response
Please find the attached file below.

Reviewer 2 Report
-page 15, panel G- “relative intensity” still needs to be better defined either on the graph or in the legend.
The authors have now included a fuller discussion regarding the role of ERK-MAPKs in AAA development. However, the last sentence of the newly inserted paragraph (lines 589-590, page 22) is clumsy as it suggests that phosphorylation of ERKs promotes AAA development, whereas I think the authors imply the converse i.e. “……PDGF-stimulated ERK1/2 activation is specifically implicated in reducing the pathogenesis of……”
Author Response
Response to the Reviewer #2 (R2)
Comment 1:
-page 15, panel G- “relative intensity” still needs to be better defined either on the graph or in the legend.
Response 1:
Thank you for your careful reading.
We corrected the label of “relative intensity” as shown in Figure 5G in page 16.
Comment 2:
The authors have now included a fuller discussion regarding the role of ERK-MAPKs in AAA development. However, the last sentence of the newly inserted paragraph (lines 589-590, page 22) is clumsy as it suggests that phosphorylation of ERKs promotes AAA development, whereas I think the authors imply the converse i.e. “……PDGF-stimulated ERK1/2 activation is specifically implicated in reducing the pathogenesis of……”
Response 1:
According to the reviewer’s comment, we modified the sentence as below.
“In contrast, our findings suggest that PDGF-stimulated ERK1/2 activation is specifically implicated in reducing depression-related AAA development.”
Thank you again for your mindful comments and suggestions.
All authors are very grateful to Reviewer #2 for carefully reviewed comments and suggestions. We strongly hope that this revised version will be acceptable by the Reviewer #2.
Round 3
Reviewer 1 Report
I do not find the authors' response satisfactory. Answering all of my comments with a single discussion paragraph is inadequate and unacceptable. In order to strengthen the study and reach the level acceptable for publication, the authors are requested to perform additional experiments suggested in my previous comments.